# Promises and Challenges of Predictive Blood Biomarkers for Locally Advanced Rectal Cancer Treated with Neoadjuvant Chemoradiotherapy

**DOI:** 10.3390/cells12030413

**Published:** 2023-01-26

**Authors:** Joao Victor Machado Carvalho, Valérie Dutoit, Claudia Corrò, Thibaud Koessler

**Affiliations:** 1Translational Research Center in Onco-Hematology, Department of Medicine, Faculty of Medicine, University of Geneva, 1205 Geneva, Switzerland; 2Swiss Cancer Center Léman, 1005 Lausanne, Switzerland; 3Department of Oncology, Geneva University Hospital, 1205 Geneva, Switzerland

**Keywords:** rectal cancer, CEA, liquid biopsy, ctDNA, cfDNA, miRNA, chemoradiotherapy, pCR, tumor response, TRG

## Abstract

The treatment of locally advanced rectal cancer (LARC) requires a multimodal approach combining neoadjuvant radiotherapy or chemoradiotherapy (CRT) and surgery. Predicting tumor response to CRT can guide clinical decision making and improve patient care while avoiding unnecessary toxicity and morbidity. Circulating biomarkers offer both the advantage to be easily accessed and followed over time. In recent years, biomarkers such as proteins, blood cells, or nucleic acids have been investigated for their predictive value in oncology. We conducted a comprehensive literature review with the aim to summarize the status of circulating biomarkers predicting response to CRT in LARC. Forty-nine publications, of which forty-seven full-text articles, one review and one systematic review, were retrieved. These studies evaluated circulating markers (CEA and CA 19-9), inflammatory biomarkers (CRP, albumin, and lymphocytes), hematologic markers (hemoglobin and thrombocytes), lipids and circulating nucleic acids (cell-free DNA [cfDNA], circulating tumor DNA [ctDNA], and microRNA [miRNA]). Post-CRT CEA levels had the most consistent association with tumor response, while cfDNA integrity index, MGMT promoter methylation, ERCC-1, miRNAs, and miRNA-related SNPs were identified as potential predictive markers. Although circulating biomarkers hold great promise, inconsistent results, low statistical power, and low specificity and sensibility prevent them from reliably predicting tumor response following CRT. Validation and standardization of methods and technologies are further required to confirm results.

## 1. Introduction

According to the European Society for Medical Oncology (ESMO) in 2017, rectal cancer (RC) represents 35% of colorectal cancers (CRC); in the European Union, the incidence of RC is 125,000 per year (15–20 cases/100,000 habitants per year), and the mortality is 4–10 deaths/100,000 habitants [1]. It is suggested that these numbers may rise in the future [1,2]. Locally advanced RC (LARC) corresponds to stage II or III RC and is the most prevalent stage at diagnosis in Europe [1,3].

For LARC, European and American medical oncology guidelines recommend curative surgery with total mesorectal excision (TME) and removal of the mesorectal nodes en-bloc [1,4]. To decrease the risk of local recurrence, neo-adjuvant radiotherapy (RT) or chemo-RT (CRT) is used [1]. CRT combines 45–50.4 Gy radiation in 25 fractions of 1.8–2 Gy with radio-sensitizing chemotherapy using capecitabin or fluorouracil (5-FU) for 6 weeks [1]. RT alone consists of a short-course radiotherapy (SCRT) of 25 Gy in 5 fractions of 5 Gy. Surgery takes place between 1 and 12 weeks after completion of neoadjuvant treatment [1]. It has been shown that longer waiting times do not improve local control but increase the chances of pathological complete response (pCR) [5,6]. The ideal time to surgery is currently unknown [7]. To date, CRT and SCRT with delayed surgery (at least 4 weeks after RT) are considered standard of care and interchangeable [6,8]. Treatment intensification with total neoadjuvant treatment (TNT)—adding consolidation or induction chemotherapy to standard of care—further improves distant control and pCR (up to 30%) as shown in the RAPIDO [9] and PRODIGE-23 [10] trials.

In LARC, despite neoadjuvant treatments, pCR occurs in only 6 to 39% of patients [11,12,13,14]. It means that a large proportion of the patients do not benefit from neoadjuvant treatment while experiencing side effects from it [15]. Therefore, being able to predict treatment response is of utmost importance as it can determine patients who would benefit from neoadjuvant treatment and minimize the harmful effects of it when it is not appropriate [16]. Another major goal is to avoid removing the rectum and losing the organ, losing its function, and introducing the associated surgical complications [15]. For patients not to be operated on, they need to achieve clinical complete response (cCR) or near cCR which could be complemented with local excision. Patients in cCR usually enter a “watch and wait” program with regular clinical, radiological, and endoscopical assessments [16].

Despite the existence of well-known measurable circulating biomarkers, none are available in clinical use to predict the tumor response to neoadjuvant treatment in LARC patients [1]. Our review aims at clarifying the performance and clinical utility of the circulating biomarkers for predicting the tumor response in the context of LARC patients undergoing CRT (Figure 1).

## 2. Materials and Methods

We performed a systematic literature search of the Embase database on 18 November 2021, using the following keywords: “locally advanced rectal cancer”, “rectal cancer”, “blood biomarker”, “biological marker”, “tumor marker”, “tumor response”, “cancer regression”, “complete response”,” preoperative treatment”, “neoadjuvant”, and “chemoradiotherapy” with the Boolean AND/OR.

We only included studies evaluating tumor response as an endpoint based on the pCR, tumor regression grade (TRG), or tumor downstaging. Studies evaluating overall survival (OS) and/or disease-free survival (DFS) only were not included. Patients needed to have stage II or III RC treated with CRT (CRT) using capecitabine or 5-FU chemotherapy and a total radiation dosage of 45 to 50.4 Gy. Notably, other long-course CRT with RT schemes of 39.6 to 63 Gy were also included (Appendix A).

When possible, sensitivity (Sn), specificity (Sp), positive predictive value (PPV), and negative predictive value (NPV) were calculated based on the cut-off values available.

## 3. Results

### 3.1. Literature Search Results

Taken together, the literature search resulted in the selection of fifteen full text articles. In addition to these studies, thirty-four articles were added after analyzing the cited literature from the fifteen selected articles. All added articles met the inclusion criteria. In total forty-nine studies were included. An overview of the selection process is provided in Figure 2.

Twenty-five of these studies investigated the following biomarkers: carcinoembryonic antigen (CEA), carbohydrate antigen (CA 19-9), thrombocytes, hemoglobin (Hb), leukocytes, lymphocytes, c-reactive protein (CRP), albumin, and lipids, while twenty-four studies focused on acid nucleic liquid biopsies, analyzing cell free DNA (cfDNA), circulating tumor DNA (ctDNA), and microRNA (miRNA) (Appendix A).

### 3.2. Protein Tumor Markers

#### 3.2.1. CEA

CEA is a glycoprotein expressed on the apical part of the normal epithelial cell membrane. In tumor cells, CEA loses its polarization, which increases its expression and concentration in the circulation [17]. CEA is a prognostic marker in CRC and is used to detect tumor relapse after surgery [1] and progression during treatment [18]. No specific cut-off value exists for local recurrence in RC [19], and its predictive value for tumor relapse pre-surgery is controversial [20]. We identified twenty-four studies investigating CEA levels as a predictor of tumor regression at different time points: pre-CRT and post-CRT (Table 1).

Pre-CRT CEA levels. Twenty-one studies investigated pre-CRT CEA levels using different cut-offs: 2.5 ng/mL [21], 2.7 ng/mL [22], 2.85 ng/mL [23], 3.5 ng/mL [24], 4.4 ng/mL [25], 5 ng/mL [26,27,28,29,30,31,32,33,34,35,36,37,38,39], and 6 ng/mL [40]. Eleven studies found that CEA values lower than the cut-off in pre-CRT was an independent and positive predictive factor of tumor response to CRT in univariate analyses [21,22,23,25,26,28,30,31,34,35,41]. In multivariate analyses, low CEA value pre-CRT remained an independent predictive factor of tumor regression in eight studies [25,26,28,30,31,34,35,41]. The ten other studies found no statistical correlation between pre-CRT CEA level and tumor regression [24,27,29,32,33,36,37,38,39,40].

Post-CRT pre-surgery levels. Ten studies explored post-CRT but pre-surgery CEA levels with different cut-offs: 2 ng/mL [32], 2.45 ng/mL [23], 2.61 ng/mL [40], 2.7 ng/mL [24], and 5 ng/mL [29,31,36,39,42,43]. Seven studies found that CEA value lower than the cut-off in the post-CRT setting is predictive of tumor regression [23,24,31,32,39,40,42] in univariate and multivariate analysis, and three studies reported no association [29,36,43].

Two studies [23,39] considered the ratio between pre- and post-CRT CEA levels. Cai et al. [39] evaluated a post-CRT/pre-CRT CEA ratio, whereas Song et al. [23] evaluated a pre-CRT/post-CRT ratio. Cai et al. [39] use 0.23 as a cut-off value and demonstrated its predictive value for tumor regression in univariate and multivariate analyses for patients with values lower than 0.23. Song et al. [23] found that a value lower than 1.07 was an independent predictive factor of pCR in univariate and multivariate analyses.

To reduce the effect that tumor size can have on CEA levels, Gago et al. [33] consider the ratio between the pre-operative CEA level and the maximum tumor diameter (measured by MRI). Using a cut-off value of 2.429 ng per mL per cm, they demonstrated that patients having a lower value after neoadjuvant CRT had a higher chance of reaching pCR (66.3% versus 14.3%), but it was not significant in multivariate analysis.

Finally, Hu et al. [43] measured CEA levels at several time points: baseline (pre-CRT) and then at 2, 6, and 12 weeks after the start of treatment. They measured the CEA clearance pattern by drawing exponential curves based on trend lines. Then, they calculated the R2 value which represent the “correlation coefficient between the trend line illustrating the exponential decrease and the measured CEA values” [43]. If the R2 is equal or close to 1, it means that the patient had an adequate tumor response, and, in fact, they find that the clearance pattern with and exponential decrease in CEA during treatment (R2 ≧ 0.9) is predictive of pCR in univariate and multivariate analysis in a cohort of 146 patients derived from the two arms of the prospective trial FOWARC [44] comparing standard CRT to modified FOLFOX6 with or without RT.

#### 3.2.2. CA 19-9

CA 19-9 is a cell surface glycoprotein complex secreted by a variety of secretory cells including pancreatic, biliary ductal, gastric, colon, endometrial, and salivary cells as part of mucous secretions [45,46]. CA 19-9 is prognostic factor in pancreatic adenocarcinoma [47,48]. It can be falsely negative in patients Lewis (Le) negative blood group, which represent 7–10% of the global population [48].

In RC, seven studies measured CA 19-9 levels using cut-off values of 3.5 U/mL [22], 9 U/mL [26], 10 U/mL [26], 12.6 U/mL [23], 35 U/mL [34], or 37 U/mL [28,30,39]. None of these studies showed significant results using CA 19-9 levels in pre- or post-CRT settings (Appendix A). When evaluating tumor downstaging, two studies showed that pre-CRT CA 19-9 level were predictive of response in univariate analysis [23,26], but only Yeo et al. [26] observed a significant association in multivariate analysis.

Yeo et al. [26] studied the predictive value of tumor downstaging for post-CRT CA 19.9 but failed to show a significant association. They evaluated the pre- and post-CA 19-9 ratio using a cut-off value of 1.28 (for downstaging) and 0.92 (for pCR). They found that lower values were predictive of tumor downstaging and pCR in univariate analyses, whereas this held true in multivariate analyses for downstaging only.

Finally, Yang et al. [30] looked at the predictive value of a decreased pre- and post-CRT CA 19.9 ratio using a cut-off value of 37 U/mL but did not observe association with tumor response.

### 3.3. Hematological Markers

#### 3.3.1. Thrombocytes

It is thought that thrombocytes can promote cancer through several mechanisms: shielding tumor cells from the immune system, supporting tumor cell extravasation, and stimulating angiogenesis through the secretion of proangiogenic cytokines such as VEGF or PDGF [49]. In turn, tumor cells can promote thrombocytosis by secreting thrombopoietin, which is correlated with poorer outcome in solid cancers such as pancreatic adenocarcinoma [50] and non-small cell lung cancer [51]. In the context of LARC, four studies looked at the predictive value of thrombocyte level in pre-CRT or post-CRT with cut-off values ranging from 253 G/L to 370 G/L [22,28,41,52] (Table 2). Only two studies demonstrated that high pre-CRT thrombocyte levels were predictive of poor likelihood of response in univariate analyses [22,28], whereas only one study remained significant for predicting pCR after multivariate analyses [28]. A single study looked at post-CRT thrombocyte levels with no significant result [41].

#### 3.3.2. Hemoglobin (Hb)

Anemia is associated with poor outcomes in several cancers [53] probably due to intratumoral hypoxia [54], which in turn stimulates angiogenesis at the tumor site via secretion of HIF and VEGF [55]. Hypoxia can further modify tumor cell metabolism and induce cell quiescence, which can increase treatment resistance, especially in the context of RT [56,57]. In LARC, ten studies explored association of pre-CRT Hb levels with tumor response after CRT [22,24,25,26,29,30,32,41,52,58]; one study considered post-CRT Hb levels [41], and one considered Hb variation during treatment [32]. Several cut-off values are investigated, including 9 g/dL [30], 10 g/dL [29,32], 12 g/dL [52,58], 12.2 g/dL [22], 12.5 g/dL [26], 13.2 g/dL [25], and 13.5 g/dL [24] (Appendix A). Regarding pre-CRT Hb level, three studies [22,41,58] demonstrate an association with tumor response in univariate analysis, but none demonstrate an association with tumor response in multivariate analysis. Post-CRT Hb levels were not associated with tumor response in any of the studies. Finally, Huang et al. showed that patients without anemia (Hb > 10 g/dL) during CRT were more likely to achieve pCR compared to those with anemia; these results remained significant in multivariate analyses [32].

### 3.4. Leukocytes and Inflammatory Markers

#### 3.4.1. Albumin and C-Reactive Protein

Inflammation is a well-known hallmark of cancer [59]. It is measured in peripheral blood using c-reactive protein (CRP), albumin, or circulating leukocytes levels [60]. We found ten studies looking at inflammatory markers as predictive markers in LARC patients treated with CRT [22,25,27,31,32,38,41,52,61,62] (Appendix A). Three studies looked at CRP or albumin [22,41,52]; four studies measured leukocytes levels [25,27,32,41]; and seven investigated the ratio between leukocytes and/or CRP and/or albumin [22,31,38,41,52,61,62] or its combinations [38].

Albumin is the most abundant serum protein [63]. Its level decreases during an inflammatory event, this being usually associated with poor outcomes [64,65]. Two studies [41,52] evaluating the predictive value of albumin in LARC patients failed to show any association of pre- or post-CRT albumin levels with tumor response.

CRP is an acute-phase protein whose concentration in plasma rises during inflammation and is routinely measured as a diagnostic marker for inflammatory events [66,67]. High CRP levels are associated with poor outcomes [68,69]. In LARC, Aires et al. [22] found that pre-treatment CRP level below 3.5 mg/L was an independent predictive factor of tumor regression in univariate and multivariate analyses.

CRP and albumin have been combined in the Glasgow prognostic score (GPS) and the modified GPS (mGPS). GPS scores from 0 to 2; one score is given for CRP levels above 10 mg/L or albumin levels below 35 g/L [70,71]. In the mGPS, isolated hypoalbuminemia (<35 g/L) does not score [72]; therefore, a mGPS score of 1 represents an isolated elevation of CRP (>10 mg/L), and a mGPS score of 2 represents an elevation of CRP with hypoalbuminemia [73]. Dreyer et al. [62] showed that a pre-CRT mGPS score of 0 was associated with increased tumor regression and predicted TRG in univariate and in multivariate analyses.

#### 3.4.2. Leukocytes

Eight studies investigated circulating leukocyte levels as a predictor of tumor response after CRT in LARC [25,27,31,32,38,41,52,61] (Appendix A). Three studies evaluated leukocyte counts before [25], during [32], or before and after [41] CRT treatment, but none showed significant association with tumor response. On the contrary, Kitayama et al. [27] showed that high lymphocyte counts pre-CRT was significantly associated with response in univariate and multivariate analysis. Likewise, Heo et al. [25] found that a high-sustained lymphocyte count after 4 weeks of CRT compared to the pre-CRT count was predictive of pCR in univariate and multivariate analyses. Noticeably, results at other time points (after 8 or 12 weeks) were not significant (Appendix A).

The neutrophil to lymphocyte ratio (NLR) reflects the dynamic interaction between innate and adaptative immunity in inflammatory events such as sepsis or cancer [74]. NLR is a prognostic marker in several solid tumor types with high ratios being associated with poorer outcomes (e.g., reduced overall survival or disease-free survival) [75,76]. In LARC, seven studies evaluated its predictive value [22,31,38,41,52,61,62], all measuring NLR pre-CRT with a cut-off ranging from 1.7 to 5 (Appendix A). In univariate analysis, two studies reached statistical significance [31,41]. One identified NLR < 2.8 associated with pCR in multivariate analysis [31]. No association could be demonstrated for NLR in post-CRT [41].

The neutrophils to albumin ratio (NAR) is predictive of mortality or has a prognostic value in several tumor types [77,78,79]. In LARC, two studies measured its predictive value for tumor response to CRT [38,41]. Out of the two studies, only Tawfik et al. [41] found that an elevated pre-CRT NAR was associated with decreased likelihood of pCR in univariate and multivariate analysis, while post-CRT NAR was not.

The lymphocyte to monocyte ratio (LMR) is associated with worse tumor control in breast or lung cancer [80,81]. Three studies investigated LMR pre- or post-CRT in LARC [38,41,61], but none showed significant results in prediction of treatment response.

The platelet to lymphocyte ratio (PLR) has been shown to be predictive of worse overall survival (OS) in several solid tumors [82]. Regarding the response to CRT in LARC, none of four studies investigating PLR showed a significant association between PLR measured pre- or post-CRT and response [31,41,61,62].

Finally, Sawada et al. [38] evaluated seven other ratios and combinations: the lymphocyte to CRP ratio (LCR), neutrophil and CRP product (N×C), monocyte and CRP product (M×C), neutrophil and monocyte product (N×M), monocyte to albumin ratio (MAR), CRP to Albumin ratio (CAR), and the prognostic nutritional index (PNI) which is calculated by using an equation combining serum albumin levels and the lymphocyte count [83]. They show that pre-CRT high LCR and low (N×M) correlate with better TRG in univariate and multivariate analysis.

### 3.5. Lipid Marker

#### Apolipoprotein A-1

Lipids can promote oncogenesis by allowing tumor cells to meet their increased metabolic demand, by modulating being part of tumor-modulating pathways, by inducing the recruitment of inflammatory cells, and by playing immuno-modulatory roles [84,85,86]. One study [34] looked at the predictive value of different lipids in relation to tumor regression following CRT. Only pre-treatment apolipoprotein A–I levels ≤ 1.20 g/L correlated with a lower proportion of responders, the association remaining significant in multivariate analysis.

### 3.6. Nucleic Acids Marker

The liquid biopsy technique is the analysis of non-solid tissues such as blood or other bodily fluids such as saliva, cerebrospinal fluid, or urine [87]. Analyzed components of liquid biopsies such as circulating tumor cells (CTC) [88] or nucleic acids (cfDNA, ctDNA, or miRNA) show great potential for clinical application in oncology for tumor early diagnosis, detection of tumor relapse after surgery, or identification of treatment targets [89]. We identified twenty-two studies evaluating cfDNA, ctDNA, or miRNA as predictor of the tumor response after neoadjuvant treatment in LARC patients. Out of twenty-two studies, twenty-one are discussed in two reviews [90,91]. 

#### 3.6.1. Cell Free DNA

The uncovering of cfDNA dates back to 1948 when Mandel and Metais et al. observed presence of DNA and RNA in the blood of healthy and diseased patients [92]. cfDNA is now detected under physiological conditions (e.g., physical activity) or pathological conditions such auto-immune, inflammatory, or oncologic diseases [93,94]. cfDNA is either released following tissue damage [95,96] or is actively secreted [97]. In the bloodstream, its half-life is 4 to 30 min [98]. cfDNA is detectable in different bodily fluids such as urine or blood [99]. In healthy individuals, cfDNA blood concentration ranges from 1 to 10 ng/mL [94].

Five studies evaluated the predictive value of cfDNA in LARC [100,101,102,103,104] with four reporting significant results (Table 3). Zitt et al. [100] showed a significant association between cfDNA levels and tumor downstaging following CRT with cfDNA decrease after surgery significantly associated with tumor response in the multivariate analyses. Using cfDNA integrity index [105]—the ratio between long and short cfDNA fragments—Agostini et al. [101] found that a lower cfDNA integrity index post-CRT was associated with increased tumor response in multivariate analysis. Sun et al. [102] showed that long cfDNA fragments (≈400 bp) abundancy and its ratio to short cfDNA fragments (100 bp) in pre-CRT were significantly associated with tumor response. They also showed that higher MGMT promoter methylation pre-CRT predicted better TRG. Similarly, Shalaby et al. [104] studied the pre-CRT methylation of MGMT and ERCC-1 promoters, with MGMT and ERCC-1 playing an important role in DNA repair mechanisms. In both cases, a hypermethylation of MGMT and ERCC-1 promoter was associated with decreased tumor regression.

#### 3.6.2. Circulating Tumor DNA

In 1989, Stroun et al. discovered the presence of ctDNA as part of the cfDNA [106]. ctDNA detection in blood is achieved using targeted or untargeted approaches [89,107]. Targeted approaches are PCR-based and require prior knowledge of specific mutation(s) of interest that will be researched for in the blood [107]. Untargeted approaches use next generation sequencing (NGS) to detect unknown mutations than can thereafter be followed [107]. This method has the advantage of not requiring knowledge of mutations before performing the test but is less sensitive than the targeted approach [107].

Nine studies evaluated ctDNA as predictor of tumor response following CRT in LARC [37,108,109,110,111,112,113,114,115]. Eight studies [37,108,109,110,111,112,113,115] considered ctDNA levels; one study used BRAF and KRAS mutations in ctDNA [113]; and one used neuropeptide Y (NPY) methylation status in ctDNA [114] (Table 4). The detection rate of ctDNA in these studies ranged from 20.5% to 75% in pre-CRT, 8.3% to 22.3% in post-CRT, and 6.7% to 13% in post-surgery samples (Table 4). Two studies showed significant results regarding post-CRT DNA levels. Khakoo et al. [112] and Zhou et al. [110] demonstrated that post-CRT ctDNA levels was associated with TRG (mrTRG) [112] or pCR [110] and TRG (CAP) [110]. Finally, Murahashi et al. [37] showed that a loss of >80% between post-CRT ctDNA and baseline ctDNA (pre-CRT) predicted TRG in univariate and multivariate analyses.

#### 3.6.3. MicroRNA

microRNAs (miRNAs) are non-coding RNA molecules which regulate gene expression and cell-cell interactions in autocrine, paracrine, or endocrine ways [116,117]. In general, onco-miRNAs are overexpressed in cancer cells and increase degradation of tumor suppressor messenger RNA (mRNA), while tumor suppressive miRNAs are usually less expressed in tumor patients [118,119,120]. In blood, miRNAs are also shed in exosomes in all bodily fluids, and recent studies suggest that they are associated with tumorigenesis [120]. Exosomal miRNAs are resistant against degradation which makes them potential biomarkers of interest [120,121].

Several studies investigating several different kinds of miRNA of pre-treatment formalin-fixed and paraffin embedded (FFPE) biopsy samples demonstrated an association with tumor response in LARC patients undergoing CRT (pCR/TRG) [122,123,124]. Eight studies investigated blood circulating miRNA predictive value of tumor response following CRT in RC [36,125,126,127,128,129,130,131] (Table 5).

Yu et al. [127] identified low pre-CRT levels of miR-345 as a predictor of TRG (Mandard). D’Angelo et al. [126] showed that pre-CRT miR-125b was preferentially overexpressed in non-responders. Hiyoshi et al. [131] showed that low pre-CRT levels of miR-43 were predictive of response to CRT; however, they did not reproduce Yu’s or D’Angelo’s results regarding miR-125b and miR-345. Azizian et al. [128] showed that reduced miR-20a and miR-18b during CRT was predictor of negative postoperative nodal status; however, no correlation with TRG was found. Wada et al. [36] identified a panel of 8 pre-CRT miRNA (miR-30e-5p, miR-33a-5p, miR-130a-5p, miR-210-3p, miR-214-3p, miR-320a, miR-338-3p, and miR-1260a) for which variations were predictors of tumor response (TRG) to CRT in LARC. Looking specifically at exosomal miRNAs, Baek et al. [130] identified pre-CRT miR-199b-5p upregulation as predictor of response in multivariate analyses. Finally, Dreussi et al. [125] investigated a series of 114 single nucleotide polymorphisms (SNPs) related to miRNA and found that DROSHA-rs10719, SMAD3-rs17228212, SMAD3 rs744910, and SMAD3-rs745103 SNPs were associated with non-pCR, whereas TRBP-rs6088619 mutation was associated with pCR.

## 4. Discussion

This review explores the potential value of several circulating biomarkers to predict LARC regression following CRT. The forty-nine selected studies looked at twelve single biomarkers (CEA, CA 19-9, Hb, thrombocytes, leukocytes, lymphocytes, albumin, CRP, apoA1, cfDNA, ctDNA, and selected miRNA) and twelve multiple biomarkers (mGPS, NLR, NAR, LMR, PLR, LCR, MAR, CAR, PNI, N×C, M×C, and N×M).

Pre-treatment (pre-CRT) is the most relevant time point when predicting neoadjuvant treatment effect as it can guide the clinician in choosing the right approach prior any treatment is administrated. In this setting, CEA is the most studied circulating marker. It shows significant association in univariate and multivariate analyses with tumor response in eight out of twenty-one studies. The most widely used cut-off is 5 ng/mL. With this cut-off, sensitivity ranges from 61.7% to 92.6%, and specificity ranges from 41.8% to 63%. These characteristics do not allow CEA to be a clinically relevant biomarker, and it is therefore not currently used to guide clinical decision. A second circulating biomarker associated with poor tumor response to CRT is detection in cfDNA of MGMT promoter hypermethylation [102,104]. However, these results originate from two small studies and warrant replication in a larger cohort of patients. Other markers studied during the pre-CRT window show either inconsistent results (e.g., CA 19-9, thrombocytes), no association (e.g., hemoglobin, albumin, or ctDNA) or positive association (e.g., CRP, apolipoprotein A-I, or miR-125b). The latest were shown in only one study and failed to be reproduced.

Single liquid biomarkers tested at a post-CRT pre-surgery time point are more consistently associated with tumor response. Some markers such as ctDNA are associated with tumor response only when measured during this period [110,112]. This observation illustrates the importance of selecting the right time point of analysis for each biomarker. Out of all biomarkers tested, CEA showed frequent and significant association with tumor response in eight studies out of eleven. Improvement in the test sensitivity, ranging from 90.9% to 100% while specificity ranged from 25% to 32.5% at a cut-off set at 5 ng/mL, was observed when this time point is used. However, this time point (post-CRT but pre-surgery) is of less clinical relevance, as neo-adjuvant CRT has already been given. It could, however, complement clinical, radiological, and endoscopic assessments for patients on a watch and wait track. Nevertheless, the true added value of these biomarkers in the watch and wait setting must be specifically studied, as none of the studies reviewed addresses this topic.

To improve on the predicting value of circulating biomarkers, authors used two different strategies: testing at multiple time points to assess biomarker change over time and ratio or testing of multiple biomarkers. For change over time and ratios measurement, usually one time point before CRT and one time point before surgery is selected. Authors use different mathematical analyses, ratios between post- and pre-CRT levels being most commonly used. Most of these analyses showed no predictive impact, with the noticeable exception of Song et al. [23] showing that CEA and CA19-9 pre-CRT over post-CRT ratios were independently and statistically associated with downstaging in a large patient cohort (*n* = 674). On the other hand, most authors exploring multiple markers were equally unsuccessful, apart from Sawada’s neutrophil × monocytes value and lymphocytes to CRP ratio [38]. Significant associations with the tumor response have also been shown by Dreussi et al. [125] using 6 SNPs and Wada et al. [36] using a panel of 8 miRNAs. However, despite a large sample size [125] or validation cohort [36], these finding have currently never been replicated. Wada et al. [36] showed that the predictive value of a miRNA panel is improved when combined with the CEA levels, highlighting the fact that multiple markers should be used as they interrogate different tumor components. Indeed, a possible explanation for the failure of other miRNA studies is that miRNA transcriptional profile is modified by the tumor microenvironment and other parameters such as the oxygenation levels [132], which CEA is less sensitive to.

Several factors might be responsible for the lack of a predictive value from the studied circulating biomarkers. First, most studies have small sample size, with on average 184 patients per study, which leaves 92 patients in each group based on a dichotomous outcome—pCR or not pCR. Limited sample size may be the most important factor as it limits statistical power to detect moderate effects. Second, variations in the pre-analytic setting (preparation prior to sampling, sampling collection method, sample conservation method, time between sampling and measurement, etc.) may hinder the comparison of the findings between different studies. Numerous studies reviewed here are based on patient records with no possibility to assess the used technic [24,41]. Third, variations in the determination of the cut-off values are present between the studies due to the lack of established cut-off value. This variability may be increased by the fact that some used technologies are still in development (e.g., ctDNA and cfDNA [133,134]) with lack of standardization leading to inconsistencies in their threshold. All these variations can lead to either false negative or positive results. Fourth, for authors using multiple time points, the time at which the second value is measured, post-CRT but pre-surgery, is of critical importance as biomarker clearance dynamic post-CRT is not standardized and, in many cases, not studied [135]. Fifth, heterogeneity in the treatments administrated in these studies can be observed, with variations in radiotherapy (total dose, boost technic, and dose per fraction) and chemotherapy (dosage) schedule, as well as time to surgery, with all these factors being known to influence pCR rate [5,136]. Sixth, confounding factors such as gender, ethnicity, age, concomitant disease, or treatment may influence the measurement of the circulating biomarkers and may induce biases in the results. This a possible explanation for some discrepancies between univariate and multivariate analyses [22,33]. Furthermore, the presence of multivariate analyses does not guarantee that all the confounding factors have been taken into account. Lastly, there is a possible heterogenicity in the tumor response assessment in the listed studies as some use a pathological score (pCR) while other use radiological downstaging. This discrepancy into the assessment method and score can lead to discrepancies between the studies [137].

This review faces several limitations. First, as it is based on studies retrieved the EMBASE dataset, it is possible that studies have been missed during the selection process.

Second, this review focuses on CRT, which is the most common neoadjuvant treatment used. However, other treatment regimens such as short-course radiotherapy (SCRT) or total neoadjuvant therapy (TNT) exist and possibly have different impacts on the tumor or its microenvironment and therefore a different impact on circulating biomarkers. Biomarker studies using these different treatments are soon to be available due to the high interest for TNT and its increased rate of pCR (up to 30%) and for SCRT due to its convenience for patients and similar outcome compared to CRT.

Currently, no single circulating biomarker is able to guide clinical decision. Multimarker models such as the Glasgow Prognostic Score (albumin and CRP) seem to have some potential at predicting tumor response to neoadjuvant CRT [62,138]. Furthermore, others circulating biomarkers (e.g., long non-coding RNA, circulating RNA, or methylated DNA) deserve to be studied [139,140,141].

Ultimately, the prediction accuracy of these models may be improved by the addition of other types of biomarkers associated with the tumor characteristics, tumor microenvironment composition, or the microbiota which could influence it [142,143,144,145,146]. The Immunoscore^®^—the combination of CD3 and CD8 T-cell densities in the tumor and its invasive margin [147]—was positively correlated with the degree of histologic response after neoadjuvant CRT [148]. More recently Chatila et al. conducted the largest (*n* = 738) genomic and transcriptomic study looking at determinants of tumor response to neoadjuvant therapy in rectal cancer [149]. If no somatic alterations had significant associations the tumor response, overexpression of IGF2 and L1CAM was associated with decreased response. Furthermore, Chatila et al. discovered a subset of microsatellite-stable tumors with an immune hot transcriptomic profile and increased response [149].

## 5. Conclusions

Although circulating biomarkers hold great promises, inconsistent results, low statistical power, and low specificity and sensibility prevent them from reliably predicting tumor response to CRT. Further validation and standardization of methods and technologies are required to confirm early results. Ultimately, a multimarker model incorporating circulating as well as tissue (tumor and microenvironment) biomarkers seems to hold the highest promise for the future clinical involvement of circulating biomarker for LARC patients.

## Figures and Tables

**Figure 1 cells-12-00413-f001:**
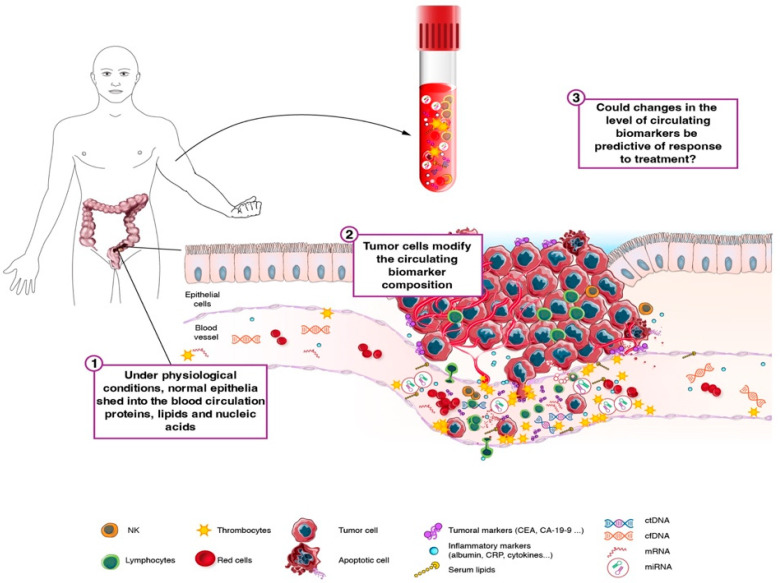
Biomarker production under physiological and pathological conditions. 1. Under physiological conditions, normal epithelia and stroma shed proteins, lipids, and nucleic acids into the blood circulation. 2. Tumors modify the circulating biomarker composition by shedding tumor markers and by modulating the normal circulating biomarker production. 3. Circulating biomarkers levels can be measured in peripheral blood, and it may be predictive of tumor response. Legend: CA 19-9: cancer antigen 19-9, CEA: carcinoembryonic antigen, cfDNA: cell free DNA, ctDNA: circulating tumor DNA, CRP: c-reactive protein, mRNA: messenger RNA, miRNA: micro-RNA, NK: natural killer cells.

**Figure 2 cells-12-00413-f002:**
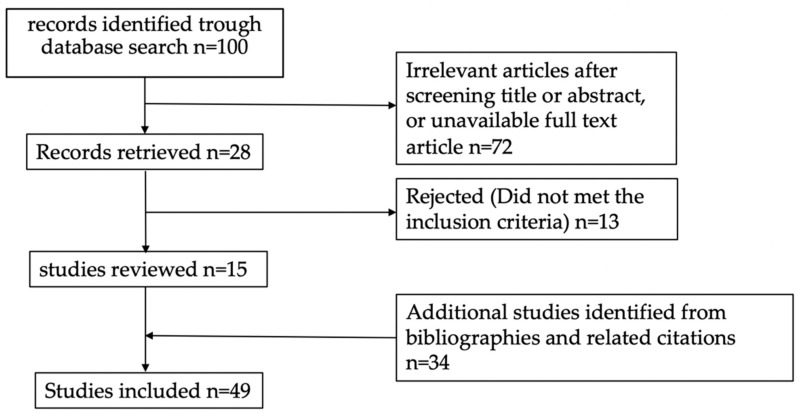
Flow chart illustrating the screening and selection process.

**Table 1 cells-12-00413-t001:** Studies exploring CEA as a biomarker predictive of tumor regression following CRT.

Study	Cut-Off	N	Measured Outcome	Sn	Sp	VPP	VPN	*p* Value (Univariate Analysis)
pre-CRT								
Engel et al. [21]	2.5 ng/mL	209	pCR	59.5%	65.3%	30.1%	86.5%	0.004
Aires et al. [22]	2.7 ng/mL	171	TRG 0–1 (Rayan)	43.6%	75.4%	n/a	n/a	0.0213
Song et al. [23]	2.85 ng/mL	674	pCR	52.2%	66.5%	28.3%	84.6%	<0.0001
Jang et al. [24]	3.5 ng/mL	109	TRG 3–4 (Dworak)	66.6%	53.1%	50%	69.4%	ns
Heo et al. [25]	4.4 ng/mL	52	pCR	14.3%	36.8%	7.7%	53.8%	<0.01 *
Yeo et al. [26]	5 ng/mL	260	Yp Stage 0–1	80.8%	45.8%	45.8%	80.8%	<0.01 *
Kitayama et al. [27]	5 ng/mL	73	pCR	70%	56.5%	20.6%	92.1%	ns
Kim et al. [28]	5 ng/mL	314	TRG 3–4 (Dworak)	81.1%	41.8%	16.4%	94%	0.007 *
Choi et al. [29]	5 ng/mL	53	pCR	63.6%	45.3%	23.3%	86.4%	ns
Yang J. et al. [30]	5 ng/mL	531	pCR	66%	50%	23.5%	86.4%	0.021 *
Cheong et al. [31]	5 ng/mL	145	pCR	92.6%	63%	38.5%	97.1%	<0.001*
Huang et al. [32]	5 ng/mL	236	pCR	71.4%	42.2%	27.8%	82.6%	ns
Gago et al. [33]	5 ng/mL	89	pCR	63.2%	43.5%	25.5%	79.4%	ns
Guo et al. [34]	5 ng/mL	751	TRG 1–2 (Mandard)	61.7%	51.8%	54.1%	59.4%	0.009 *
Zhang et al. [35]	5 ng/mL	432	pCR	n/a	n/a	n/a	n/a	0.001 *
Wada et al. [36]	5 ng/mL	106	pCR	72.2%	30.1%	18.3%	83.3%	ns
Murahashi et al. [37]	5 ng/mL	85	pCR	n/a	n/a	n/a	n/a	ns
Sawada et al. [38]	5 ng/mL	267	TRG 3–4 (Dworak)	n/a	n/a	n/a	n/a	ns
Cai et al. [39]	5 ng/mL	284	pCR	n/a	n/a	n/a	n/a	ns
Yang K.L. et al. [40]	6 ng/mL	138	pCR	n/a	n/a	n/a	n/a	ns
Tawfik et al. [41]	n/a	98	pCR	n/a	n/a	n/a	n/a	0.002 *
post-CRT pre-surgery								
Huang et al. [32]	2 ng/mL	236	pCR	16.1 %	93.9%	45%	78.2%	0.0285 *
Song et al. [23]	2.45 ng/mL	674	pCR	66.9%	54.5%	27.1%	86.7%	<0.0001 *
Yang K.L. et al. [40]	2.61 ng/mL	138	pCR	76%	58.4%	n/a	n/a	0.026 *
Jang et al. [24]	2.7 ng/mL	109	TRG 3–4 (Dworak)	88.9%	42.2%	51.9%	84.4%	<0.001 *
Choi et al. [29]	5 ng/mL	53	pCR	90.9%	14.3%	22.2%	100%	ns
Cheong et al. [31]	5 ng/mL	135	pCR	100%	25%	25%	100%	0.008 *
Wada et al. [36]	5 ng/mL	106	pCR	94.7%	4.9%	18.8%	80%	ns
Cai et al. [39]	5 ng/mL	284	TRG 0–1 (NCCN)	n/a	n/a	n/a	n/a	<0.001 *
Restivo et al. [42]	5 ng/mL	260	pCR	95.3%	32.3%	21.8%	97.2%	<0.0001 *
Hu et al. [43]	5 ng/mL	71	pCR	81.8%	36.7%	19.1%	91.7%	ns
post-CRT/pre-CRT ratio								
Cai et al. [39]	0.23	284	TRG 0–1 (NCCN)	n/a	n/a	n/a	n/a	<0.001 *
pre-CRT/post-CRT ratio								
Song et al. [23]	1.07	674	pCR	38.2%	73.8%	26.9%	82.5%	0.006 *
clearance pattern (R^2^)								
Hu et al. [43]	0.9	71	pCR	81.8%	63.3%	29%	95%	0.008 *
pre-CRT/tumor size ratio								
Gago et al. [33]	2.4 ng/mL per cm	89	pCR	82.4%	19.6%	23.7%	78.6%	0.04

*: also significant in multivariate analysis. Legend: CRT: chemoradiotherapy, NCCN: national comprehensive cancer network, NPV: negative predictive value, ns: non-significant, n/a: not available, N: number of patients, pCR: pathological complete response, yp Stage: pathological stage after neo-adjuvant treatment, PPV: positive predictive value, Sn: sensitivity, Sp: specificity, TRG: tumor regression grade.

**Table 2 cells-12-00413-t002:** Studies exploring thrombocytes level as a biomarker predictive of tumor regression following CRT.

Study	Cut-Off	N	Measure Outcome	Sn	Sp	VPP	VPN	*p*. Value (Univariate)
pre-CRT								
Kim et al. [28]	370 G/L	314	pCR	94.7%	22.8%	14.6%	96.9%	0.01 *
Krauthamer et al. [52]	350 G/L	140	pCR	68.2%	42.9%	34.9%	75%	ns
Aires et al. [22]	253.5 G/L	171	TRG 0–1 (Rayan)	75.5%	47.8%	n/a	n/a	0.0018
Tawfik et al. [41]	n/a	98	pCR	n/a	n/a	n/a	n/a	ns
post-CRT pre-Surgery								
Tawfik et al. [41]	n/a	98	pCR	n/a	n/a	n/a	n/a	ns

*: also significant in multivariate analysis. Legend: CRT: chemoradiotherapy, NPV: negative predictive value, ns: non-significant, n/a: not available, N: number of patients, pCR: pathological complete response, PPV: positive predictive value, Sn: sensitivity, Sp: specificity, TRG: tumor regression grade.

**Table 3 cells-12-00413-t003:** Studies exploring cfDNA as a biomarker predictive of tumor regression following CRT.

Study	Method	N	Measure Outcome	Measured Marker	Time Point	Significant Markers (*p* < 0.05)
Zitt et al. [100]	qPCR	26	ypDownstaging	cfDNA levels	pre-CRT post-CRT post-Surgery	post-Surgery cfDNA levels
Agostini et al. [101]	qPCR	67	TRG (Mandard)	cfDNA levels Integrity index	pre-CRT post-CRT	post-Integrity index
Sun et al. [102]	qPCR	34	TRG (Dworak)	cfDNA levels cfDNA integrity MGMT promoter methylation KRAS mutation	pre-CRT post-CRT	pre-CRT 400 bp cfDNA concentration cfDNA integrity pre-CRT MGMT promoter methylation
Schou et al. [103]	Direct Fluorescence	123	pCR	cfDNA levels	pre-CRT post-CRT	ns
Shalaby et al. [104]	qPCR	93	TRG (Dworak)	MGMT and ERCC-1 promoter methylation	pre-CRT	pre-CRT methylation of MGMT and ERCC-1 promoters

Legend: cfDNA: cell free DNA, CRT: chemoradiotherapy, ns: non-significant, N: number of patients, pCR: pathological complete response, qPCR: quantitative polymerase chain reaction, TRG: tumor regression grade.

**Table 4 cells-12-00413-t004:** Studies exploring ctDNA as a biomarker predictive of tumor regression following CRT.

Study	Method	N	Measure Outcome	Measured Marker	Time Point (Detection Rate)	Significant Markers (*p* < 0.05)
Murahashi et al. [37]	Amplicon-based deep sequencing	85	TRG (Dworak)	ctDNA level	pre-CRT (57.6%) post-CRT (22.3%) post-Surgery	ctDNA reduction (post-CRT/pre-CRT ctDNA levels)
Pazdirek et al. [108]	singleplex PCR	36	TRG (Dworak)	ctDNA level	pre-CRT (21.2%) during-CRT	ns
Carpinetti et al. [109]	WGS	4	TRG (Dworak)	ctDNA level	pre-CRT during-CRT post-CRT	ns
Zhou et al. [110]	NGS	104	pCR TRG (CAP)	ctDNA level	pre-CRT (75%) during-CRT (15.6%) post-CRT (10.5%) post-Surgery (6.7%)	post-CRT ctDNA level
McDuff et al. [111]	NGS ddPCR	29	pCR	ctDNA level	pre-CRT post-CRT	ns
Khakoo et al. [112]	ddPCR	47	mrTRG	ctDNA level	pre-CRT (74%) during CRT (21%) post-CRT (21%) post-Surgery (13%)	post-CRT ctDNA Level
Sclafani et al. [113]	ddPCR	97	CR (RECIST 1.1)	ctDNA level KRAS/BRAF mutation	pre-CRT (50–66%)	ns
Appelt et al. [114]	ddPCR	146	TRG (Mandard)	Meth-ctDNA (NPY)	pre-CRT (20.5%)	ns
Tie et al. [115]	NGS	159	pCR	ctDNA level	pre-CRT (77%) post-CRT (8.3%) post-Surgery (12%)	ns

Legend: CRT: chemoradiotherapy, ctDNA: circulating tumor DNA, CR: complete response, CAP: college of american pathologists, ddPCR: dropplet digital polymerase chain reaction, mrTRG: magnetic resonance tumor regression grade, meth-ctDNA: methylated circulating tumor DNA, NGS: next-generation sequencing, N: number of patients, pCR: pathological complete response, PCR: polymerase chain reaction, TRG: tumor regression grade, WGS: whole-genome sequencing.

**Table 5 cells-12-00413-t005:** Studies exploring miRNA as a biomarker predictive of tumor regression following CRT.

Study	Method	N	V	Measure Outcome	Measured Marker	Time Point	Significant Markers (*p* < 0.05)
Wada et al. [36]	qRT-PCR	41	65	TRG (Mandard)	miRNA (8-panel)	pre-CRT	All panel: miR-30e-5p miR-33a-5p miR-130a-5p miR-210-3p miR-214-3p miR-320a miR-338-3p miR-1260a
Dreussi et al. [125]	DNA sequencing	265	/	pCR	miRNA-related SNP (114-panel)	n/a	*DROSHA*-rs10719 *SMAD3*-rs17228212 *SMAD3* rs744910 *SMAD3*-rs745103 *TRBP*-rs6088619
D’Angelo et al. [126]	qRT-PCR	34	/	TRG (Mandard)	miRNA (11-panel)	pre-CRT	miR-125b
Yu et al. [127]	qRT-PCR	87	42	TRG (Mandard)	miRNA (16-panel)	pre-CRT	miR-345
Azizian et al. [128]	Real time PCR	42	/	Lymph Node Negativity	miRNA (5-panel)	pre-CRT during CRT post-CRT	miR-20a miR-18b
Meltzer et al. [129]	n/a	29	64	TRG (CAP)	Exosomal miRNA (372 panel)	pre-CRT	ns
Baek et al. [130]	qRT-PCR	89	/	TRG (Rödel)	Exosomal miRNA (16-panel)	pre-CRT	Exosomal miR-199b-5p
Hiyoshi et al. [131]	RT-PCR	94	/	TRG (Dworak)	miRNA (18-panel)	pre-CRT	miR-43

Legend: CRT: chemoradiotherapy, CAP: college of american pathologists, miRNA: micro-RNA, n/a: not available, N: number of patients, V: number of patients in a validation cohort, qRT-PCR: quantitative real time polymerase chain reaction, pCR: pathological complete response, RT-PCR: real time polymerase chain reaction, SNP: single nucleotide polymorphisme, TRG: tumor regression grade.

## Data Availability

Not applicable.

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
