# Peer review of "Promises and Challenges of Predictive Blood Biomarkers for Locally Advanced Rectal Cancer Treated with Neoadjuvant Chemoradiotherapy"

_cells, 2023, doi:10.3390/cells12030413_

Round 1

Reviewer 1 Report (Previous Reviewer 1)

After revision, I suggest that this article can be accepted and published in the journal of "Cells". 

Reviewer 2 Report (Previous Reviewer 2)

Dear Authors, 

I hope my suggestions have helped improve your review, and I think it will be of interest to readers as it stands.

This manuscript is a resubmission of an earlier submission. The following is a list of the peer review reports and author responses from that submission.

Round 1

Reviewer 1 Report

Overall Comments:

Dr. Carvalho et al. Have submitted a manuscript reporting the outcomes of “Promises and challenges of predictive blood biomarkers for locally advanced rectal cancer treated with neoadjuvant chemoradiotherapy”. The authors performed a systematic literature search of database and concluded that pre-CRT CEA levels had the most consistent association with tumor response. This study is a very well written study and presents findings in a clear and easy to read manner. But I have minor revised suggestion to the authors as follows:

Minor revision:

1.In the Figure 2, the rejected numbers were 12 or 13? Because the rejected number was 12 in the figure but the numbers of the studies reviewed were 15. Please correct it.

2.After analyses of the including articles, the authors found that pre-CEA level has the most consistent association with tumor response for these LARC patients with neoadjuvant CCRT. But the cut-off value of CEA pre-CRT in these articles is consistent? For example, the last CEA value before CRT. In addition, if the pre-CRT/post-CRT CEA ratio is more consistent association with tumor response than the pre-CRT CEA level?

Reviewer 2 Report

The authors collected and summarized scientific data on prognostic blood biomarkers used to assess and/or predict the response of locally-advanced colorectal cancer (LARC) to neo-adjuvant chemoradiotherapy (CRT).

The review looks comprehensive, but when reading it, the question arises: why does this clinical aspect need to be specially analyzed?  Is there any reason why a particular marker might be more or less appropriate for predicting response to neo-adjuvant versus adjuvant therapy, for monitoring locally advanced versus advanced cancer? The same list of markers appears to be applicable to various clinical aspects of colon cancer and has recently been reviewed: https://doi.org/10.2147/CMAR.S253369, https://doi.org/10.1186/s12943-022-01556-2; https://doi.org/10.1038/s41416-022-01769-8; https://doi.org/10.3389/fcell.2021.660924). Authors need to point out why certain markers are required to predict LARC response to CRT and what is the main advantage of this review over others.

The authors have compiled a large list of reports on CEA. However, this marker has long been well studied. I would be interested to know the opinion of the authors, why twenty-four studies give such different diagnostic values for this parameter?

Moreover, several types of circulating marker were left out of attention: cyrcular RNA (doi: 10.1111/jcmm.16380); long non-coding RNA (doi: 10.3748/wjg.v25.i34.5026), exosomes (doi: 10.3390/biomedicines9080931), methylated DNA would require more attention (doi: 10.3390/cimb43030100).

Thus, it is recommended to clearly define the purpose of this review (1), to condense information about well-established markers (2), and to supplement the review with at least a mention of new markers (3).

Reviewer 3 Report

Review of the manuscript entitled « Promises and challenges of predictive blood biomarkers for locally advanced rectal cancer treated with neoadjuvant chemoradiotherapy” by Joao Victor Machado Carvalho et al

This review aims to clarify the performance of predictive circulating biomarkers for correctly classifying those patients with rectal cancer who underwent chemoradiotherapy in terms of local response. This goal is necessary since more than 30% of them may not need surgery after such therapies.

Thus, CA 19-9 : cancer antigen 19-9, CEA : carcinoembryonic antigen, cfDNA : cell free DNA, ctDNA : circulating tumor DNA, CRP : c-reactive protein, mRNA : messenger RNA, miRNA : micro-RNA, NK : natural killer cells are described and not putted in perspective with gene, metabolomic or immune cell markers in tissues.

Methods used is based on systematic literature search of the Embase database focusing on articles published  with reference to pCR, tumor regression grade (TRG), or tumor downstaging, evaluating OS and/or DFS.

The review allows summarizing the following messages

1-      They do not conclude regarding CA 19-9; they give a summary of cut-off values through different studies for CEA without any comments on the methods used for blood collection, conservation, and efficiency of antibody used; in brief how comparing no centralized tests; CEA to be a clinically relevant biomarker and it is therefore not currently used to guide clinical decision

2-      CRP and albumin have been combined in the Glasgow prognostic score (GPS) and the modified GPS(mGPS). They take as a fact that pre-CRT mGPS score of 0 was associated with increased tumor regression and predicted TRG in univariate and in multivariate analyses.

3-      In a study, they found that a high sustained lymphocyte count after 4 weeks of CRT compared to the pre-CRT count was predictive of pCR in univariate and multivariate analyses. Because lymphocyte and leukocyte ratios may be predictive of innate and adaptative response to tumours, In LARC, seven studies evaluated its predictive value all measuring NLR pre-CRT with a cut-off ranging from 1.7 to 5; One study identified NLR < 2.8 was associated with pCR after multivariate analysis and no one showed interest of NLR post-CRT.

4-      In one study, pre-CRT high LCR (CRP/Ly ratio) and low (NxM; neutrophil and monocyte) correlate with better TRG in univariate and multivariate analysis

5-      Pre-treatment apolipoprotein A-I levels ≤ 1.20 g/L correlated with a lower proportion of responders, the association remaining significant in multivariate analysis.

6-      Pre-CRT methylation of MGMT and ERCC-1 promoters, MGMT and ERCC-1 play an important role in DNA repair mechanisms. In both cases, an hypermethylation of MGMT and ERCC-1 promoter was associated with decreased tumor regression

7-      Regarding circulating DNA, a significant (>80%) loss between post-CRT ctDNA and baseline ctDNA (pre-CRT) predicted TRG in univariate and multivariate analyses.

8-      miRNA levels have been reviewed without any clear-cut selection since there are multiple markers and time to recovery is not consensually established

9-      Thus, the authors statute that multiple markers should be used as they interrogate different tumor components. They attempt to explain the interaction between various biological parameters; as an example, they insist  on a possible explanation for the failure of other miRNA studies that miRNA transcriptional profile might be modified by the tumor microenvironment and other parameters such as the oxygenation levels, which CEA is less sensitive to.

Major issues

Methods for metanalysis. They identified 100 articles amongst only 15 have been analyzed. Then 34 new articles have been added to this series. There is not clear whether the new series met all inclusion criteria of the study or because new keywords have been found in their preliminary research; if so, why authors did not extend their research on the Embase database by using new keywords. Nevertheless, the following issues are the most important

Although the catalog below may be intellectually useful, there are some issues with the authors’ approach. The main failure of this review is that the authors have not a critical lecture on the techniques and various tests needed for measuring these markers. Methods and technical issues are essential regarding the measurement of markers, particularly in the blood milieu (time of recovery, delay for banking, conditions under which samples are conserved until measurement etc.. . We know there is not a consensual standardized technic for measuring. Thus, they can not take any conclusion on none of these tests.

Second, three major studies (Ref 143, 144, 145) are cited in this review showing immune cell infiltrate quantification, MSI, and genomic and transcriptomic analyses as determinants of tumor response to neoadjuvant therapy; these could be relevant as alternative markers in tissue. There is no clear why tissue markers appear to be more relevant than blood markers are not more detailed. Further, all these parameters have been shown influenced by luminal and tissue-adherent microbiota, a biological component that is not at all mentioned in this study.

Third, many confounding markers such as age, co-morbidities, and medicines that are daily taken by patients may influence both levels (of systemic markers), as well as response to therapies. This bias is not discussed nor evaluated in the articles reviewed here.